# Academics' Epistemological Attitudes towards Academic Social Networks and Social Media

Jevgenija Sivoronova [1,2,*] , Aleksejs Vorobjovs [2] and Vitālijs Raščevskis [2]

[1]  Institute of Humanities and Social Sciences, Daugavpils University, LV-5401 Daugavpils, Latvia
[2]  Education and Psychology Department, Faculty of Humanities and Social Sciences, Daugavpils University, LV-5401 Daugavpils, Latvia; aleksejs.vorobjovs@du.lv (A.V.); vitalijs.rascevskis@du.lv (V.R.)
[*]  Correspondence: jevgenija.sivoronova@du.lv

**Abstract:** Academic social networks and social media have revolutionised the way individuals gather information and express themselves, particularly in academia, science, and research. Through the lens of academics, this study aims to investigate the epistemological and psychosocial aspects of these knowledge sources. The epistemological attitude model presented a framework to delve into and reflect upon the existence of knowledge sources, comprising subjective, interactional, and knowledge dimensions. One hundred and twenty-six university academics participated in this study, including lecturers and researchers from different higher education institutions in Latvia. The study employed two methods: the Epistemological Attitudes towards Sources of Knowledge Questionnaire and the Epistemological Attitudes towards Sources of Knowledge Semantic Questionnaire. The data analysis involved several procedures, including exploratory and confirmatory factor analysis, correlation analysis, and test statistics. By implementing these methods, the study gained valuable insights into the sources of knowledge, examining them from two perspectives. The first perspective brought attention to the differences in academics' appraisals by discussing their understanding, approach, use, and valuations of these sources. By scrutinising the constructs of meanings, the second perspective sheds light on the anticipated knowledge which is deemed ideal, the concrete knowledge that is both social and objective, and the subjectively valuable nature of academic social networks and social media. The findings underscore the specialised knowledge and qualities that academics rely on for producing knowledge. In terms of epistemology, methodology, social science, and education, the study holds theoretical and practical implications, especially in comprehending knowledge and its sources.

**Keywords:** epistemological attitude; knowledge sources; social media; academic social networks; university academics

## 1. Introduction

Knowledge sources continue to be a current and crucial topic in academia, presenting various possibilities for understanding and raising questions about their quality. Nowadays, a wide array of knowledge sources are available for academic, scientific, and educational purposes. These comprise scientific journal articles, monographs, books, knowledge shared by university lecturers, textbooks and handbooks, popular science magazines and books, and digital media, such as academic social networks and social media. Academics, including assistants, lecturers, senior lecturers, associate professors, professors, and researchers, are accredited as experts in cognising and creating the content of these sources.

The emergence of the informational and interactive paradigm in the present academic landscape has rendered digital media a crucial source for academic cognition, communication, and information. Academics now exploit digital media through multiple roles as proficient users and audience members, listeners, readers, communicators, and authors or creators. Social media (SM) and academic social networks (ASNs) are platforms that

facilitate knowledge acquisition, creation, and sharing through various forms of communication, including academic, scientific, educational, or personal dissemination [1]. The actuality of cognition holds a central position in both platforms. An active presence on SM apprises one of current events, promotes engagement in discussions on critical social issues, and aids in staying informed of scientific discourses and discoveries. Moreover, an active presence also enhances the likelihood of conversing with individuals sharing similar interests [2]. Essentially, individuals gain knowledge and engage as active creators on digital media platforms.

In light of the importance and various considerations surrounding the sources, we have formulated our primary research question: What are the epistemological and psychosocial aspects of ASNs and SM from an academic perspective? This primary question comprises two sub-questions that provide further clarification. The first sub-question explores how academics cognise, approach, use, and value ASNs and SM as reliable sources of knowledge. The second sub-question aims to establish the semantic constructs of these knowledge sources and investigate their epistemological and psychosocial meanings.

This article is structured into six sections to provide a comprehensive understanding. The second section discusses the shared characteristics and distinctions between ASNs and SM. Further, the third section delves into the theoretical framework appertaining to epistemological attitudes. The fourth section outlines the methodology and process employed in this study. Following that, the fifth section presents the results that throw light on the research questions and provides interpretations. Finally, the article is concluded in the sixth section with a discussion.

## 2. Academic Social Networks and Social Media: Common Features and Differences

Despite the shared purpose of serving as sources of knowledge, marked dissimilarities have been observed between the two platforms. These differences encompass a range of cognitive possibilities, sociodemographic characteristics of users and authors, the nature of communication, and the knowledge content. The distinctions can be perceived while considering the wide array of content, services, and tools these sources provide [3].

### 2.1. Academic Social Networks

ASNs are potent interactive platforms that academics strategically employ for both professional and personal use. Among the noteworthy ASNs are ResearchGate, Academia.edu, Kudos, Mendeley, PhilPapers, Common Ground Research, ImpactStory, Google Scholar, and other similar scholarly media platforms. These ASNs provide five primary services to support scholarly work. They help scholars disseminate their research to a wider audience, manage their online identities, offer metrics to measure academic impact, facilitate the search for relevant scientific communities, and assist in managing a vast amount of information, comprising references, literature, and documents. Additionally, platforms like ResearchGate, Academia.edu, LinkedIn, Mendeley, and ImpactStory offer copious features such as discussion boards, file repositories, messaging services, citation counts, public or semi-public profiles, group collaboration options, collaborative document processing, network visibility, publication uploads, and integration with other social media platforms [3].

ASNs significantly impact various aspects of the academic experience and social and professional communication [4]. They enable academics to share their work [5,6], connect with peers and prospective collaborators [7,8], establish social networks, and promote themselves [9,10]. Furthermore, these platforms allow scholars to assert their individuality within the academic community [11]. The factors discussed above imply ASNs' role in shaping the social capital within academia, accentuating the interrelation between ASNs' metrics and academic factors like position, rank, research field, and institutional rank [4]. Social capital refers to the tangible and intangible resources that individuals or groups acquire from building and maintaining a network of relationships based on mutual acquaintance and recognition [12]. The impetus for using ASNs and strengthening

social capital include self-promotion and increasing self-esteem, professional knowledge acquisition, affiliation to a peer community, and interaction with peers [13].

The significance of social capital is demonstrated in the popularity of ASNs among academics. This relevance can be observed in ASNs' use as communication tools by educational establishments like higher education and research institutions, thereby enhancing social capital. Academic social platforms offer extensive information, foster meaningful interactions with like-minded individuals, and facilitate performance tracking [14]. These platforms have shifted from being solely informative to adopting a more conversational and dialogic communication model [15]. At present, numerous universities and research institutes are officially on academic and SM platforms to facilitate knowledge exchange. Additionally, these platforms raise awareness and dispense knowledge regarding education and research institutions. They enable affiliates, academics, and interested users to view updates and posts from institutions and share and comment on them. This communication boosts exposure and expands the outreach of institutions, nurturing correspondence among academics [2].

*2.2. Social Media*

Unlike ASNs, which are primarily focused on the academic field, SM is a collection of diverse web-based platforms that enable the sharing of information through content creation in various fields [16]. LinkedIn, X (commonly known by its former name, Twitter), Telegram, Facebook, Instagram, Reddit, YouTube, and other similar social networking sites are most notable among these platforms. Users can create personal profiles, connect with friends, and engage in social interactions on these platforms. This includes uploading, liking, and commenting on content such as photos, messages, and videos shared on newsfeeds [17]. SM posting performance refers to the frequency and activity on social networks, involving designing and developing a strong online presence [15]. It allows users to create and manage content collaboratively for specific purposes [18]. Similar to ASNs, SM platforms allow individuals to express their uniqueness to a wider audience [19].

According to a recent report, 57.8% of working-age internet users use SM and the internet in general to connect with friends and family, and stay updated on news and current events. Around 38% of users also use SM for educational and study purposes, while approximately 28% use it for business-related activities [20]. Using SM for work significantly impacts on work efficiency, suggesting that practitioners can improve their efficiency by using SM as a workplace tool [21]. This can also be linked to the concept of ASNs.

The influence of SM extends to individuals at both local and global levels. SM use has become increasingly widespread and common, resulting in positive and negative outcomes [22]. Studies have reported the adverse effects of SM, such as increased distractions and disturbances [23], as well as depressive symptoms [24]. The excessive use of SM has been linked to various negative consequences, including decreased productivity [25]. Various negative consequences of SM on user autonomy have also been reported. SM can control user data, attention, and behaviour, which can lead to disregarding their capabilities and harm their autonomy competencies. This includes impairing their ability to independently select content, influencing their desires, distorting their true, practical identities, and limiting critical reflection [26]. The consequences of being addicted to SM, such as developing personal identity and image disorders [27], are more severe. Such addictions occur when individuals become dependent on seeking attention and validation, feeling a strong urge to conform to the social standards aggressively promoted on SM platforms. As a result, a polarisation of attitudes towards SM can be observed in society [28], which is reasonable.

Comparing oneself in SM [24] can trigger cognitive processes that later extend to using other media content, resulting in negative effects. The "comparison-based system" in academic publishing, which encourages scholars to publish their work based on this

system and incurs substantial expenses to maintain the reputation of high-quality journals, is a current topic of debate among researchers [29,30].

The impact of SM on public opinion is unquestionably immense. SM use various tactics to manipulate and present social knowledge, ultimately influencing the expected behaviour of their users. Current studies indicate that it is crucial to consider multiple SM platforms as sources for integrating SM data. This integration can enhance the persuasiveness and comprehensiveness of online public opinion analyses. It is particularly significant owing to the harmful effects of misinformation circulating within SM networks, which tremendously affects the development of online public opinions [31]. Such adverse effects are evidence for the requirement of SM platforms to possess inventive mechanisms that compel users to assume responsibility for the information they provide and disseminate, directing them towards social knowledge that fosters social capital rather than perpetuating misinformation, non-constructive social comparison models, and the devaluation of intellectual pursuits in favour of non-sustainable business models.

While the use of SM has notable disadvantages, SM are also acknowledged for their advantages, such as engaging in civic participation [32] and enhancing social capital [33]. Positive outcomes of social capital include increased self-esteem, involvement in social life, and life satisfaction [34]. Akin to ASNs, communication through SM platforms like Facebook, X, Instagram, and Snapchat is built on the foundation of frequent users, establishing bridging and bonding social capital. The former retains the potential for distant and fragile connections between individuals to form a platform for exchanging knowledge. In contrast, bonding social capital on SM platforms entails close relationships that lend emotional connection, trust, and social aid [35].

The capacity of SM to assist in communication and the development of relationships, as well as the sharing of experiences and content creation, has been previously acknowledged [36]. This provides insight into the motivations for using SM, comprising cognition, interaction, and creativity. These motivations encompass exchanging information, conforming to social norms, expressing oneself freely, maintaining social connections, and engaging in recreational activities, where the strongest predictor of SM use is the desire to forge new connections [37]. Therefore, the primary functions of SM are to fulfil primary needs such as socialisation, entertainment, information, self-status seeking [38], self-expression, and intimate connections [39]. SM are also linked to six primary sources of reward, including passing the time, showing affection, following fashion, sharing problems, demonstrating sociability, and enhancing social knowledge [40].

Like ASNs, SM are effective channels for accomplishing strategic communication objectives [41] that are effectively exploited by higher education institutions. The effectiveness of universities' communication strategies relies on their presence and activity on SM, and the tools they use for interaction and content sharing [1,42,43]. The advantages of using SM like Facebook, LinkedIn, and X for institutional communication have been highlighted, particularly regarding posting performance, interactivity focus, and content combination. These features offer professionals a strategic guide for managing communication and engagement with individuals expressing interest [44]. By leveraging SM, universities can foster relationships with the community [1] and emphasise their brand [45]. Content quality also plays a critical role in SM communication strategy, as it helps establish the institution's position on social networks [44,46]. However, content quality not only has functional implications but also must hold substantial value, such as knowledge.

### 2.3. Knowledge Issue

The main content SM and ASNs provide is knowledge, which directly influences individuals' use of these platforms. The content varies on both platforms, and is determined by the audience and users, as well as by their intentional and unintentional cognition. Content creation can be purposeful or spontaneous, catering to cognitive and communication purposes. The cited platforms also emphasise the significance of knowledge creation and exchange.

SM are advantageous for knowledge transfer owing to their dynamic and flexible nature. They have the potential to foster a diverse and inclusive workforce, facilitating knowledge sharing within and outside of institutions. Furthermore, the findings support the effectiveness of SM in disseminating important information, such as vision, goals, performance, and welfare concerns within institutions [23]. These roles of SM highlight their key functional aspects: content production and distribution, conveying information, and using it for official and personal purposes [47].

Similar findings are observed for ASNs, where researchers primarily use them to acquire information, share information to a lesser extent, and rarely interact with others, as compared to the interactions observed with SM [13]. Additionally, the perceived credibility of ASNs' content impacts the intention to use it continuously. Conversely, the perceived effectiveness, perceived effort, and perceived social support influence an individual's systematic and continuous use of ASNs, with less impact from conscious intention [48]. This suggests that the credibility of ASNs is an essential factor in organising deliberate and targeted knowledge acquisition.

The role of digital media in facilitating knowledge acquisition and communication is the primary focus of numerous studies. Meanwhile, open issues regarding the content and epistemological characteristics of ASNs and SM in terms of knowledge provided and their quality still exist. Epistemological aspects play a key role in distinguishing these platforms. Scientific criteria such truth value, internal validity, credibility, applicability, external validity, transferability, consistency, reliability, dependability, neutrality, objectivity, and confirmability [49] can be applied to ASNs, given their scholarly and pedagogical content. However, applying these criteria to SM is a more complex matter. Furthermore, the negative impact of SM on social knowledge is often indirect and implicit, making it challenging to investigate the root cause of such negative aspects. This lacuna calls for a systematic approach at multiple levels of analysis.

Another point to consider is the complexity of the knowledge issue, where objective facets of knowledge interact dynamically with users as authors and personalities. This implies that the content of ASN and SM sources contains both existing social and objective knowledge, and the subjective knowledge of individual authors and creators. The impetus for the presentation and creation of knowledge is variable, resulting in alternative forms of knowledge. Individuals can be motivated by various factors when engaging in knowledge production, interpretation, and mediation activities. As a result, throughout the processes of producing, transferring, and acquiring knowledge, an individual assumes multiple roles, functioning not only as an author and user, but also as a creator and publisher, thus presenting three fundamental domains of knowledge concurrently. The domains comprise the information or knowledge itself as a source, which serves as both an instrument and a product. Furthermore, they involve the author's personality and motivation, and the social context in which cognition occurs.

The theoretical framework we propose aims to describe and explain the various domains of knowledge through the lens of the cognition system theory and its epistemological attitude (EA) concept [50]. Applying the EA concept will enable us to analyse the epistemological and sociopsychological aspects of knowledge. Hence, this can contribute to addressing the research gap in studies focusing on knowledge encompassing ASNs and SM.

## 3. The Cognition System Theory

In this essay, we introduce the theory of cognition system as a novel perspective within the holistic constructivism paradigm. We aim to provide a new angle into the understanding of cognition. Carefully developed, this theory considers the contributions of philosophy and social science, resulting in a comprehensive approach. Our approach delves into the co-construction of ontologies (the nature and being of reality) [51], epistemologies (the study of knowledge and cognition) [52], and methodologies.

To conceptualise cognition holistically, we have employed a three-level methodology. These three levels comprise philosophical, general scientific, and specific scientific

methods [53], each contributing to the development of the cognition system. Within each level, we have applied the deductive method, based on postulates or axioms, as Descartes outlined [54], to perform a vertical analysis and proceed with synthesis. Through this approach, we have formulated statements by discerning the pre-established ideas of esteemed philosophers, scientists, and thinkers as fundamental postulates and principles.

In the theory of cognition system, we have developed a conceptual apparatus comprising the concept of epistemological attitude (EA) as a central component. The EA is a systemic concept that takes on various forms, allowing us to represent, describe, and explain cognition phenomena and mechanisms, and develop a model of cognition. The EA concept also enables the construction of the cognition model as a specific action occurring within various dimensions. Figure 1 provides a detailed presentation of this theoretical elaboration, and the following three subsections will explain the logic behind this theoretical framework.

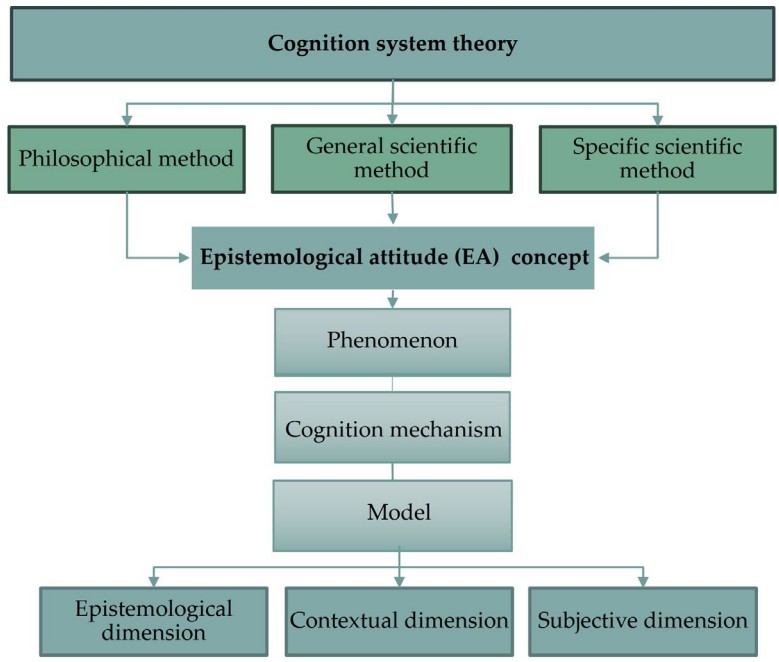

**Figure 1.** The cognition system theory.

### 3.1. Contributions of Philosophy, General Science, and Specific Sciences to the Cognition System

Following the principle of the co-construction of ontologies, cognition is shaped within the philosophical method, encompassing three forms of being. The first form is the subjective, as observed by Kant [55] and Fichte [56]. The second form is the objective, as Hartmann [57] and Bhaskar [58] expounded. The third form is the transcendental, formulated by Plato [59] and conceptualised by Kant [60]. These forms propose the existence of three fundamental beings: the subject (referred to as an agent or individual in general terms), the object or reality, and knowledge. Therefore, we advocate that cognition unfolds across these three distinct levels and involves three fundamental entities. We comprehend these levels of cognition as interconnected by defining what cognition is, who and what is involved, and how they are engaged.

Drawing on the principles of epistemological realism [61–63] and constructive epistemology [64–67], we perceive cognition as a constructive process. The results of this process are the constructions of what is observed and cognised, as well as the extractions made by the subject when presented with the observed object [68]. In this way, we consider any individual's activity as cognition, forming the basis of the subjective level of being. Then, by embracing the principles of social epistemology [69,70], we underscore the social nature of knowledge. Based on these principles, we propose that social cognition is the primary form of cognition for individuals as social beings, and we consider social mechanisms as central

processes in knowledge formation. Social epistemology framework helps us comprehend the objective level of being. By applying the principles of constructivism [71], we highlight two fundamental aspects regarding the subjective and objective levels of being. Firstly, we recognise that any activity is constructive in nature. Secondly, we assert that an individual's consciousness and reality can be expressed as knowledge. Additionally, we connect these constructivist principles to phenomenology, particularly the work of Husserl [72]. By examining more complex levels of individual activity, such as consciousness and reflection, we shed light on how the subject and reality manifest as knowledge. This position implicitly refers to the transcendental level of being. The term "transcendental" is associated with Kant's philosophy of transcendental idealism, which emphasises the primacy of knowledge. Hence, transcendental explanations are a fundamental basis for deriving other principles (a priori knowledge) [60] about the entire cognition system.

To elucidate the interconnectedness of these three levels of cognition, we have employed the philosophical category of relation. Relation, first defined by Aristotle as a means of comprehension of how one entity is connected to another [73], serves as the fundamental category for the ontology of the cognition system. This category encompasses various patterns of manifestation, such as action and interaction, as highlighted by Kant [60], as well as attitude and mechanism [74,75], enabling its application to the epistemology and methodology of the cognition system. Thus, relation can be understood in three different ways: ontologically, as a form of being, when entities relate; epistemologically, as cognition, when entities act and interact; and epistemologically and methodologically, as an attitude and mechanism, when explaining how cognition functions.

The general scientific method for modelling and studying cognition comprises systems philosophy [76], systems approach [77], and systems principles [78–80]. The system is a fundamental principle of cognition, organising the central concepts of the philosophical method into a cohesive system. We comprehend cognition as the system of relations between subsystems, which are organised into three ontologies and their entities: the subject, reality, and knowledge. The general system principle also encompasses other principles that explain how cognition functions and develops. These principles include self-organising and implementing mechanisms, which involve translating various forms of knowledge and influences, coordinating subsystems, and making probabilistic predictions regarding the behaviour and actions of each subsystem and within the system. Moreover, one of the pivotal principles establishes the presence of an integrative attribute within the cognition system, which is a systemic quality. The systemic quality enables us to conceptualise meta-relation as further elaborated by the contributions of specific sciences and move closer to conceptualising the EA.

The specific scientific method comprises concepts, theories, methods, and procedures that are applied in social science, epistemology, and education. This apparatus is used to define the principles of activity and functioning of each cognition system's entity. It explains how individuals act, interact with reality and objects, acquire and create knowledge, and how the systemic quality as meta-relation provides the functioning of various knowledge levels. The subjective constructivism [68,81–84] and social constructivism [85–89] approaches and theories reflect the activity levels engaged in the subject, object, and knowledge relations. Theories on personality activity, such as the activity principles of personality [53,90], personalisation theory [91,92], activity theory [93], and personality motivation in field theory [81], are employed to emphasise the subject as a source of activity and elucidate the underlying factors driving activity. In addition, we apply the principles of interactionism [94,95], pragmatism, and functionalism [96,97] to attribute these principles to cognition. This indicates that knowledge creation, use, and transfer involve active participation, practical application, and operational procedures at various levels of activity.

To elucidate meta-relation, we employ the principles of Gestalt theory. According to this theory, the interaction between at least two different systems leads to the emergence of new qualities, generally referred to as phenomena [98]. Therefore, we define meta-relation as the cognition phenomena that arise from various multidimensional relationships and the

reflection of "information" encompassing the entire cognition system. This definition highlights two crucial aspects: first, that cognition functioning is manifested in the phenomenal realm or the realm of knowledge, and, second, that these processes occur at various levels.

An illustrative example would be the implicit relationships that influence individuals' cognition and knowledge during interactions with reality, objects, and other individuals, which, in turn, are influenced by said individuals. These implicit relationships contribute to the development of intra-individual or subjective knowledge, as well as inter- or social knowledge. Other relations encompass the connections between reality and knowledge, including tangible and intangible things, and objective and social knowledge, which shape society, culture, and science.

Furthermore, relations between an individual and the realm of knowledge incorporate various types of cognition with different complexity. These activity levels go beyond mental activity and behaviour and encompass reflection, pure intuition, and transcendental cognition, as studied and described, for example, by Kant [60] and Husserl [72]. These advanced levels of activities contribute to the development of all forms of knowledge. Transcendental knowledge, along with subjective and objective knowledge, is one of these forms. When we use the term "transcendental", we refer to philosophical knowledge encompassing all possible knowledge, ways of cognition, and actions from all time perspectives. The term originates from the Latin words "transcendere", with "trans" meaning "across", "over", or "beyond", and "scandere" meaning "climb". It denotes something superior and beyond our comprehension that cannot be explained scientifically. It is independent and separate from our experiences [99] (p. 315). This knowledge is also recognised as meta-knowledge, which relates to Plato's realm of ideas [100] and Rugg's knowledge pyramid [101,102]. We provide Figure 2 as a visual aid to enhance the comprehension of the cognition system, meta-relation, and the EA as an integral part of meta-relation, representing the phenomenological and functional aspects of cognition. The forthcoming paragraphs will delve deeper into the EA concept and model, providing conceptual means for studying cognition.

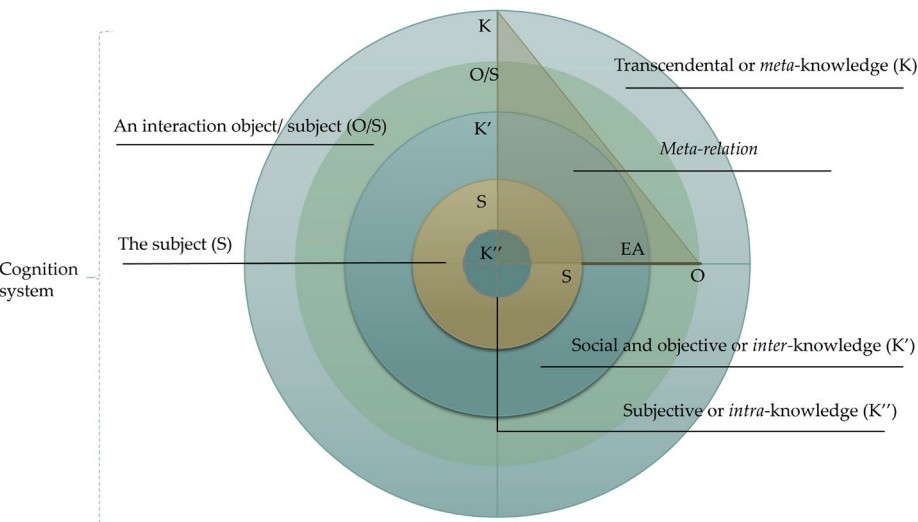

**Figure 2.** The cognition system model.

### 3.2. The Concept of Epistemological Attitude

Epistemological attitude (EA) refers to an epistemological relation within the cognition system, as shown in Figure 2. This relation is recognised as a fundamental relationship between the subject and object, originating from ancient philosophy, particularly from the teaching of Socrates, and later solidified as fundamental by Descartes [54]. Our main emphasis in developing the framework for the EA has been on epistemological relation, aiming to conceptualise and model it for integration into empirical research. According to the philosophical method, we affirm that the category of relation can take on multiple forms,

including action, interaction, attitude, and mechanism. Consequently, we comprehend epistemological relation as encompassing all these mentioned forms.

Within the framework of the general science method, epistemological relation can be recognised as a systemic part of meta-relation. Therefore, it is also considered a cognition phenomenon that plays a crucial role in the creation and transmission of knowledge. We aim to emphasise the framework of knowledge conceptualising the EA, and among the various forms of epistemological relation, attitude is regarded as the most suitable for this objective. Furthermore, in social cognition theory, the concept of attitude is recognised as both knowledge and action, with the latter referring to behaviours and psychological reactions [103]. This scientific interpretation not only supports our approach to conceptualising the EA but also validates the extrapolation of various forms of relation to the EA, which itself embodies these multiple forms.

The EA is defined as the interaction between the subject (an individual) and the object of reality, encompassing three levels of knowledge: subjective or intra-, social and objective or inter-, and transcendental or meta-knowledge [50]. We advocate that the EA, as part of the systemic quality of the cognition system, embraces all levels of knowledge that manifest in cognition. The EA comprise tangible interactions that involve knowledge, sustaining the cognition system. Consequently, this results in an individual's cognition, such as experience, understanding, the use and creation of knowledge, and various cognition-related phenomena.

As per the earlier premises, the EA manifests itself in three forms, each with its function. The first form is referred to as a cognition phenomenon, denoting its presence as an attribute or occurrence within the cognition system [98]. The second form is known as a cognition mechanism, which establishes, integrates, maintains, and facilitates complex relationships between individuals, reality, and knowledge to maintain cognition [75]. The third form is a model that operationalises the cognition mechanism. The EA as the cognition mechanism and its model represent cognition as the dynamic interplay (action and interaction) between an individual and an object across three realms or dimensions, corresponding to three co-ontologies and knowledge levels: subjective, contextual, and epistemological.

The subjective dimension encompasses an individual's psychological, biological, and social aspects, ultimately shaping their being and actions. The knowledge utilised, acquired, and created within it is subjective and varies according to an individual's cognitive abilities and personality characteristics. These facts pertain to one's psychological being [81,82,84]. The contextual dimension refers to the factual reality encompassing all present connections, events, and facts. It expands the subjective dimension as individuals develop relationships and interact with tangible and intangible objects and other individuals. Implemented knowledge refers to the entirety of an individual's accessible knowledge, including social knowledge and behavioural patterns, objective social knowledge (social norms and culture) [58,85,86], and objective scientific knowledge [62,64,89]. The epistemological dimension pertains to a specific realm of knowledge that encompasses understanding reality, objects, individuals, cognition, and existence. It denotes a comprehension of philosophical, metaphysical (theorising about origins) [73], and transcendental (a priori and superior) knowledge. We refer to this as meta-knowledge, as it extends beyond objective knowledge. These three dimensions interrelate and unify cognition elements in the systemic cognition process of perceiving, understanding, action, and knowledge acquisition, use, transfer, and creation. In the subsequent subsection, we will delve into the EA model, which offers a framework for studying the cognition of different objects.

### 3.3. The Epistemological Attitude Model

The EA model can capture and reflect cognition as the subject's actual interaction with any object. It posits that any object can be observed and investigated through its relations with the subject and knowledge. Therefore, implementing the EA model within the present study context can enhance our understanding of how academics engage with and respond to real-world entities, such as sources of knowledge. This also allows us to examine the

various knowledge levels involved in these interactions. In light of this, we formulated the EA content model to effectively seize the dynamic relationship between specific individuals, like academics, and various sources of knowledge. The study of epistemology and specific scientific theories has allowed us to identify the foundational concepts that govern this process of cognition. We have focused on psychological and sociological approaches to conceptualise the understanding of individuals and interactions, and epistemology for sources and their knowledge. After careful analysis and synthesis, we have identified four concepts that recognise individuals as active participants interacting with a source of knowledge. We acknowledge their interactions as the context in which they engage, and the knowledge realm as the force that shapes and enriches their interaction.

Figure 3 illustrates the EA within the cognition system, representing the subsystems involved. These subsystems comprise an academician as the subject, knowledge sources like digital media as the object, and knowledge subsystems such as intra-, inter-, and meta-knowledge. The EA model accentuates the epistemological, contextual, and subjective dimensions. Within these dimensions, the illustrated content model encompasses four concepts: epistemological strategy and epistemological approach, both belonging to the epistemological dimension, cognition context in the contextual dimension, and the activity principle of personality associated with the subjective dimension. Each concept is defined by categories that demonstrate the content and functional characteristics. In this section, we have provided the definitions of these concepts, and in the method section, we have further elaborated on the definitions of the features.

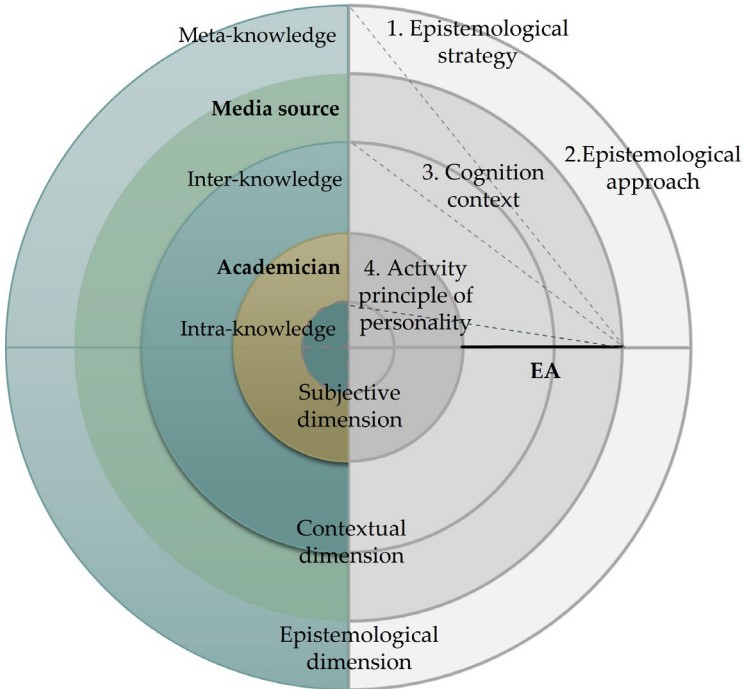

**Figure 3.** Epistemological attitude model in the cognition system (adopted from [104]).

1.  The epistemological strategy relates to an individual's overall stance on the cognisability and understanding of the world and the cognitive potential and predictability of knowledge sources and their contents. It encompasses three features: optimism (positivism) [105], scepticism [54], and agnosticism [60].
2.  The epistemological approach concerns an individual's methods for formulating and resolving knowledge issues and the specific cognitive activity involved in comprehending the sources. Eight distinct paired features exist that can be categorised into four classical and four non-classical epistemological approaches: criticism and post criticism, fundamentalism and normativism, the rejection of fundamentalism and

normativism, subject-centrism and the rejection of subject-centrism, science-centrism and the rejection of science-centrism [66,67].

3. The cognition context refers to the characteristics, frequency, and intensity of interaction implemented by an individual with the source and its application within a definite context. The classification of academic cognition entails distinguishing between academic and personal contexts.

4. The activity principle of personality pertains to the determining factors and driving forces that govern an individual's interaction with and utilisation of various knowledge sources, encompassing personality, motivation, and the pragmatic value of knowledge. This concept draws ideas from personalisation theory, principles of personality activity, philosophical pragmatism and functionalism, and non- and situational cognitive motivation theory. Three distinct cognitive motivations have been synthesised: selection [53,90], homeostasis (balance) [81,82], and reduction [96,97,104].

The EA offers a method to analyse the rationale for an individual's object selection, interaction with it, and the knowledge they engage and use. Based on this model, we have developed two instruments grounded in the methodology used to analyse meanings. Meaning refers to how the mind processes and connects concepts, objects, or phenomena and conveys the mental representation of these connections, as defined by Fodor [106]. The semantic methodology, also known as the semantic approach, method, and analysis, is employed to investigate the meanings of knowledge, phrases, representations of objects, and categories, as established by Osgood [107]. It was further developed by Petrenko [108] as a specific approach that investigates the meanings in consciousness: psychosemantics. Considering the study of knowledge and its epistemological categories and psychosocial value, it is apparent that the concept of meaning, referred to as semantics or psychosemantics, and used synonymously in this context, offers the most suitable approach to address the systemic framework and provide answers to the research questions. The ensuing methodology section introduces the empirical part of the study and provides details on the participants, the description of the instruments, and the data analysis procedure.

## 4. Methods and Materials

### 4.1. Participants and Procedure

An empirical study was conducted on a random sample of 126 academics from various higher education institutions (universities and colleges) in Latvia. The sample was made up two groups, which were determined using two questionnaires. The first group comprised 66 academics (38 females and 28 males) aged between 33 and 80 (mean age = 50.97, SD = 11.53) who consented to participate in the study. These academics had diverse scientific and academic backgrounds, with all holding at least a Master's degree: 31.8% held a Master's degree, and 68.2% had a PhD degree. Regarding their positions within the institution, the study cohort comprised 4 assistants, 12 lecturers, 21 senior lecturers, 12 associate professors, 6 professors, 7 researchers, and 4 heads of department. The academics were affiliated with various branches of science, with 34 in Social Science, 10 in Medicine and Health Science, 7 in Humanities, and 15 in Science.

For the second group, 60 academics (32 females and 28 males) aged 34 to 72 years (mean age = 53.15, SD = 10.02) agreed to participate in the study, which was conducted using the second questionnaire. Like the first group, the participants had at least a Master's degree (33.3%) or a PhD (66.7%). The academics held the following positions within the institution: 2 assistants, 16 lecturers, 25 senior lecturers, 6 associate professors, 3 professors, 2 researchers, and 2 heads of department. The affiliation with the different branches of science was as follows: 20 in Social Science, 3 in Medicine and Health Science, 27 in Humanities, and 10 in Science. The academics from both groups voluntarily assented to participation and completed the questionnaires in both online and paper formats from March to May 2023.

*4.2. Measures*

The present research used the Epistemological Attitudes towards Sources of Knowledge Questionnaire (EAQ) [104] and the Epistemological Attitude towards Sources of Knowledge Semantic Questionnaire (EASQ) [109] to investigate the EAs of academics towards sources of knowledge and their content. The EAs were determined considering two sources of knowledge: academic social networks (ASNs) and social media (SM). Both questionnaires included socio-demographic data such as gender, age, education/scientific degree, branch of science, and position, to be filled in by the respondents.

4.2.1. Epistemological Attitudes towards Sources of Knowledge Questionnaire

The EAQ comprised 99 statements, each assessing a knowledge source using the Likert five-point scale. These statements were divided into four sets according to the four domains of the EA. Each domain has its unique set of features, totalling sixteen, and a response gradation scale that varies accordingly. The response gradation scale is designed to align with the function of each domain, focusing on agreement, congruence to the source, frequency, and congruence to the respondent.

1.  The epistemological strategy domain includes three scales. An agreement response gradation scale is used to evaluate the sources, ranging from 1 (strongly disagree) to 5 (strongly agree).

    *   Optimism refers to an individual's belief in the world's complete cognisability. Such individuals have confidence in obtaining reliable knowledge from various sources and trust in the knowledge of others.
    *   Scepticism involves an individual's belief in only a partial understanding of reality. They doubt the reliability of knowledge from different sources.
    *   Agnosticism is an individual's acceptance of the fundamental unknowability of reality. They acknowledge that obtaining reliable knowledge from various sources is unattainable.

2.  The epistemological approach domain consists of eight scales. The response gradation scale measures the congruence between statements and the source. It ranges from 1 (description does not match with the source at all) to 5 (exactly describes the source).

    *   Criticism involves an individual's critical thinking, understanding, and evaluation of knowledge while considering the sources.
    *   Post criticism refers to partial criticism, where an individual accepts certain knowledge foundations and applies them in new directions beyond strict critical positions when interpreting the contents of sources.
    *   Fundamentalism and normativism represent individuals whose cognition is based on strict, established norms. They accept theoretical principles and methodologies to acquire and justify knowledge from the sources.
    *   The rejection of fundamentalism and normativism entails that individuals acknowledge the variability of norms related to cognition and the impossibility of formulating strict and stable rules for interpreting the sources that are constantly being developed.
    *   Subject-centrism refers to an individual's cognition focused on their subjective understanding. The individual's thoughts and experiences are considered the most important and reliable in acquiring and justifying knowledge.
    *   The rejection of subject-centrism is characterised by an inter-subject-oriented cognition. Individuals understand knowledge sources through communication and interaction with others. They consider both their mental processes and external reality when acquiring knowledge.
    *   Science-centrism refers to an individual's cognition influenced by the scientific context of a given time or paradigm. Such individuals rely on scientific methods and theories to comprehend reality and the sources to obtain reliable knowledge.

- The rejection of science-centrism involves acknowledging the importance of non-scientific forms of cognition. Individuals acknowledge that different sources of knowledge can be comprehended through both scientific and non-scientific perspectives.

3.  The cognition context domain comprises two scales. A gradation scale for frequency response is used to evaluate the quality of source utilisation. This scale ranges from 1 (never) to 5 (always).

    - Academic context refers to how individuals engage with various knowledge sources and apply that knowledge within an academic setting and environment in activities like science communication, educational contexts, teaching, and research.
    - Personal context involves how individuals interact with various sources and apply their knowledge in personal, interpersonal, and social situations in their everyday lives.

4.  The activity principle of the personality domain comprises three scales. A response gradation scale assesses the level of consistency between respondents' views of the source and the statements provided. The scale includes the following levels: 1 (very untrue of me), 2 (somewhat untrue of me), 3 (neither true nor untrue of me), 4 (somewhat true of me), and 5 (very true of me).

    - Selection refers to an individual's deliberate and meaningful cognitive engagement and volition in seeking knowledge from various sources.
    - Homeostasis refers to an individual's motivation to adapt functionally, encompassing the intellectual and pragmatic cognition of various sources, and the purposeful resolution of cognitive problems to achieve a mental balance between personality and the environment.
    - Reduction pertains to an individual's mental comfort and satisfaction derived from engaging with different sources of knowledge, including successfully resolving cognitive problems while minimising mental and physical resources.

All statements within each scale are summarised to obtain scores, and standardised values are calculated to compare the scales. The scores are interpreted according to the definitions of the features and their intensity.

### 4.2.2. Epistemological Attitudes towards Sources of Knowledge Semantic Questionnaire

The EASQ is composed of 92 statements. These statements are organised according to the domains and features of the EA model, representing a distinct significance for each feature. The purpose of these statements is to appraise selected sources of knowledge. Each statement is evaluated based on its semantics (meanings), which describe various aspects of the source's content, functions, and characteristics. The semantic differential scale has been accommodated to measure the semantics of these statements [107]. Each meaning is expressed as the object of the statement, through semantic categories, such as quality, function, thought, emotion, action, behaviour, value, and other mentioned criteria, which help characterise the sources and their attributes. Each statement has its own unique set of semantic categories.

The semantic differential scale, employed for evaluating each object of the statement (semantic categories), utilises a seven-point response rating scale (−3, −2, −1, 0, 1, 2, 3). The response gradation follows the pattern as mentioned: "3" is the highest rating for the object being described in the statement; "−3" signifies the complete opposite or absence of the object being described in the proposed statement; and "0" represents a rating that relies on the meanings of the poles of the response scale. These poles can include the following options: "neither agree nor disagree", "neither descriptive nor undescriptive of the source at the same time", "sometimes", "partially descriptive and undescriptive", "partially both", or "not explicitly both". The midpoints (−2, −1, 1, 2) represent the gradation between adjacent options. Consequently, each option expresses the extent to which the statement

describes the sources of knowledge, measuring the meanings. The scores of the statements have not been summarised; instead, they are considered individual items to investigate the epistemological and psychosocial meanings of the sources and to create semantic constructs that encapsulate their meanings.

Cronbach's alpha coefficient for all items of the EAQ was 0.91, indicating the questionnaire's good reliability. The EASQ also demonstrated good reliability with Cronbach's alpha coefficient of 0.89. The internal consistency of the EAQ scales ranged from $\alpha = 0.52$ to $\alpha = 0.90$. Scales such as "post criticism", "the rejection of fundamentalism and normativism", and "subject-centrism" had satisfactory $\alpha$ values (>0.65), while scales like "rejection of science-centrism" and "reduction" had inadequate $\alpha$ values (<0.60). Statistically significant correlations ($0.05 > p > 0.000$) were discerned between the scales. The reliability analysis of the EASQ scales exhibited a range of $\alpha$ from 0.57 to 0.96. However, Cronbach's alpha coefficient for the scale "the rejection of subject-centrism" was only 0.33, which can be attributed to its polar categories. The "criticism", "post criticism", and "the rejection of science-centrism" scales presented with satisfactory $\alpha$ values (>0.65). In contrast, "the rejection of fundamentalism and normativism" and "fundamentalism and normativism" scales had inadequate $\alpha$ values (<0.60). Statistically significant correlations ($0.01 > p > 0.000$) were also observed between the scales. The insufficient and low $\alpha$ indices can be attributed to the complex categories and their diverse interpretations within the scales. The process of developing and improving the EAQ and EASQ is currently underway.

The EAQ and EASQ were also subjected to exploratory factor analysis (EFA) to determine their three-factor structure. Items with low factor weights (<0.4) were excluded from the analysis. The three resultant factors were the subjective, contextual, and epistemological dimensions of the EA, with a particular focus on the specificities of the analysed knowledge sources. Although EFA also confirmed the scales, further investigations are required to enhance their reliability. To test the four-factor content model for EAQ, a confirmatory factor analysis (CFA) was further conducted. The analysis demonstrated a satisfactory fit of the model to the data: $\chi^2(83) = 478.75$, $p > 0.000$, CFI = 0.90, TLI = 0.86, RMSEA = 0.09, SRMR = 0.05. The findings support using the initial model to analyse the main research questions. However, efforts are underway to improve the model fit by reducing the number of items and increasing the indices.

*4.3. Data Analysis*

First, we analysed socio-demographic data, where descriptive statistics such as mean, standard, deviation, and percentages were calculated. Second, we analysed the EAQ data using the Friedman test to examine the EAs of academics towards each knowledge source. Furthermore, we employed the Mann–Whitney test to compare academics' EAs towards ASNs and SM. We also analysed the EASQ data to establish semantic constructs of ASNs and SM, which involved creating correlation matrixes of semantic categories (92 × 92) and performing exploratory factor analysis. The entire analysis was conducted using SPSS Statistics and Amos version 28.0.1.1.

## 5. Results

This research focuses on academics' EAs towards ASNs and SM, addressing the primary research question, and explores the epistemological and psychosocial aspects of both sources. In response to the initial sub-question regarding the cognition, approach, use, and value of internet-based knowledge sources among academics, we undertook separate analyses of the two sources and subsequently conducted a comparative analysis.

*5.1. Academic Social Networks*

The Friedman test was employed to analyse the results of academics' EAs towards ASNs. The findings showed that academics have specific orientations in comprehending, approaching, using, and valuing the source. The differences between the EAs features (scales) were significant (chi-square = 150.11, $p = 0.000$). The comparison of the features

by mean rank revealed thought-provoking tendencies. Optimism (mean rank of 9.95) and scepticism (mean rank of 9.28) stood out as academics' most exercised epistemological strategies. Optimism embodies a positive comprehension of the world, focusing on categories such as "understandable and acceptable content", "the essence of the subject", and "at the centre of current issues". On the other hand, scepticism is primarily based on the need to "verify the content" and acknowledge the "limited descriptive, explanatory potential of the content". The rejection of subject-centrism (mean rank of 11.83) emphasises the epistemological approach of academics in managing knowledge issues. This indicates that academics do not prioritise one individual prism, but rather focus on cognition within the source through "dynamic relationships with the real and current" and implementing "different approaches to the problem to investigate together". Furthermore, fundamentalism and normativism (mean rank of 9.74) were identified, referring to the source providing "fundamental knowledge for understanding the surrounding world" and its content corresponding to high normative knowledge and solid methodology. However, a tendency towards subject-centrism was also perceived, where the author's individuality is valued, particularly in terms of the "unique manner of the author's understanding". Academics prefer to interact with ASNs within the academic context (mean rank of 8.48), prioritising its "use in organisational and pedagogical work", followed by its "use in organising and implementing research and cognitive activities". According to the activity principle of personality, the motivation for homeostasis (balance) (mean rank of 11.20) significantly contributes to engagement and the understanding of ASNs and the expectant value of that knowledge. This motivational balance is evident through "an expanded view of the surrounding reality" and "guidance in the search for knowledge".

*5.2. Social Media*

The results of academics' EAs towards SM also revealed a specific orientation as evidenced by significant differences (chi-square = 391.97, $p$ = 0.000) between the features. One prominent epistemological strategy observed was scepticism (mean rank of 13.62), where academics acknowledged an incomplete understanding of reality through SM and expressed doubts about the reliability of knowledge obtained from such sources. Additionally, there was a tendency towards agnosticism (mean rank of 10.27), meaning that some academics accept the unknowability of reality through SM and consider obtaining reliable knowledge from it impossible. This strategy was highlighted through categories such as "low predictive potency", "linguistic constructions rather than knowledge", and even "content is absolutely uncertain".

The epistemological approach of academics aligned with their strategy, characterised by criticism (mean rank of 11.46), referring to the notion that "knowledge is difficult to distinguish from subjective beliefs", that "content imposes a prescribed point of view", and "radical criticism of authors'/members' prevails". Subject-centrism (mean of rank 11.41) and the rejection of subject-centrism (mean rank of 11.42) were both employed as methods to approach the content, implying that SM emphasise individuality in understanding and providing knowledge, as exemplified by user profiles and posting options. Simultaneously, SM prioritise interactions, and academics approach this platform through the lens of inter-subjectivity and creativity in the co-creation of knowledge within the actual context. Academics approaching SM also acknowledged the rejection of fundamentalism and normativism (mean of rank 10.06). They reasoned that strict and stable standards could not be established through knowledge disseminated through SM. Furthermore, academics comprehended the content by rejecting science-centrism (mean rank of 9.29). The criteria they emphasised include "addressing specific, specialised and not just scientific matters" and "selecting theories through a pragmatic and eclectic principle".

Academics' interaction with SM is primarily influenced by personal context (mean rank of 7.61) rather than academic context (mean rank of 2.89). Correspondingly, the principle of reduction (mean rank of 8.79) determines the cognition of SM. Academics prioritise motives such as "leisure" and understanding "without intellectual and moral

effort" when engaging with SM. Additionally, the driving force of homeostasis (mean rank of 7.41) is reasonably comparable to reduction. Academics have "an interest in gaining knowledge about the world", and SM provide them with "an expanded view of the surrounding reality".

### 5.3. Academic Social Networks and Social Media: Differences and Similarities

A comparative analysis revealed the diverse EAs of academics towards ASNs and SM, indicating that they perceive, approach, use, and signify knowledge from these sources differently. The Mann–Whitney test found significant differences between 13 of 16 EA features (0.05 > *p* > 0.000). Figure 4 compares the mean ranks of academics' EAs towards the sources.

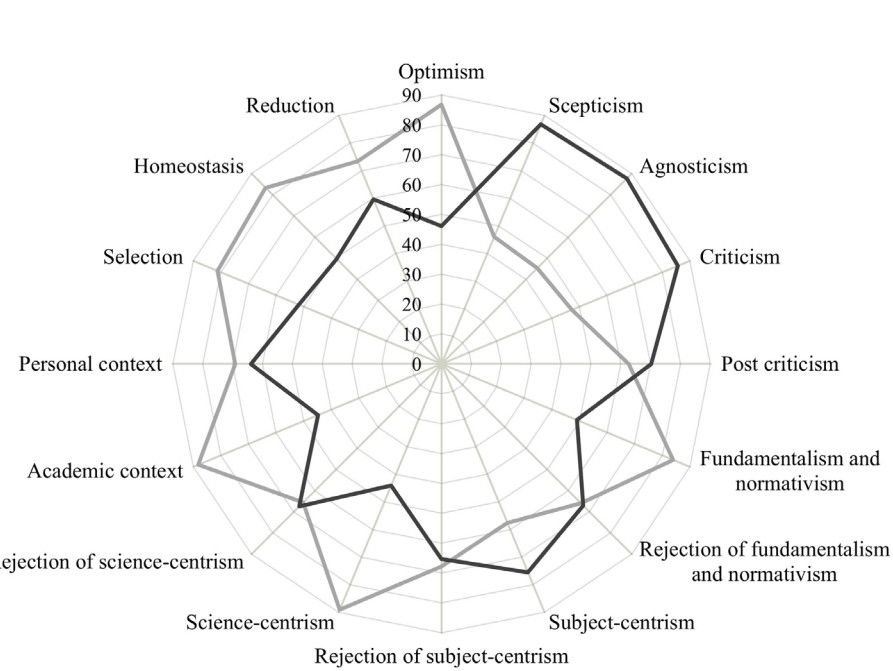

**Figure 4.** Academics' epistemological attitudes towards academic social networks and social media (the profiles of the mean ranks).

The results showed that academics have discrete epistemological strategies, meaning that they anticipate disparities in perception and differ in their comprehension of the sources. The differences were observed to be significant in optimism (U = 836, *p* = 0.000), scepticism (U = 834.5, *p* = 0.000), and agnosticism (U = 780.5, *p* = 0.000). Figure 4 demonstrates that academics express a notable optimism towards ASNs. Compared to SM, academics expect to benefit more from ASNs in terms of receiving "understandable and acceptable content", gaining insights into "the essence of the subject", obtaining "an expanded and in-depth view of the subject", and taking precedence over SM. Reasonably, both scepticism and agnosticism are more inclined towards SM. Academics resonated with the idea that SM knowledge "should be verified", may "not always offer a plausible explanation", and is more "a transformation rather than creation". They also acknowledged that content from SM "arouses interest but not trust" and that there may be "deficiencies in the proofs".

Academics have distinct epistemological approaches to the knowledge sources, employing diverse methods to formulate and resolve knowledge problems. The differences were found to be significant in criticism (U = 910, *p* = 0.000), fundamentalism and normativism (U = 1025.5, *p* = 0.000), subject-centrism (U = 1587, *p* = 0.007), and science-centrism (U = 692.50, *p* = 0.000). Academics critically approach knowledge in SM owing to their

predetermined viewpoints, knowledge-belief problem, "the susceptibility to criticism", and "the ineffectiveness of conventional problem-solving approaches". However, they are more inclined towards subject-centrism to SM than ASNs. SM content is approached in an individualised manner, involving "a method of subjective interpretation", and recognising "the author's personal theoretical perspective". In contrast, academics perceive knowledge obtained from ASNs as relying on "models, norms, and foundational knowledge". It is considered "fundamental knowledge", "based on a paradigm and theoretical methodology", and able to "distinguish between subjective and objective knowledge", with "a tendency to address global issues". Fundamentalism and normativism are supported by science-centrism, characterising ASNs as containing knowledge "acknowledged by scientific authorities", "reliable knowledge excluding subjective representations", and demonstrating "a proactive and constructive orientation in scientific thinking", following "scientific methodology" while contrasting "scientific and metaphysical knowledge". However, both sources express post criticism, the rejection of fundamentalism and normativism, the rejection of subject-centrism, and the rejection of science-centrism. No statistical differences (>0.05) were obtained, indicating a mainstream cognition of digital content and a post-classical orientation in academics' approaches to knowledge.

Significant differences were observed within the cognition context pertaining to the academic context (U = 743.50, $p$ = 0.000). It was observed that academics are more frequent users of ASNs compared to SM concerning "academic communication", "organising and implementing research", "the development of study courses or projects", and "organisational and pedagogical work". Academics rely more on "colleagues' and students' feedback" to use ASNs than SM. They also use ASNs for an "in-depth analysis" of subjects and find them helpful "for novel and extraordinary solutions in education and research". ASNs are more frequently and intensely used by academics in both contexts. However, no significant differences between ASNs and SM were perceived in the personal context ($p$ > 0.05). Academics employ and apply knowledge from both sources in personal, interpersonal, and social situations in their everyday lives. However, ASNs are predominantly regarded as "a source of inspiration and new ideas".

The activity principle of personality is crucial in determining academics' cognition. Significant differences in principles of selection (U = 1205.00, $p$ = 0.000), homeostasis (U = 1066.5, $p$ = 0.000), and reduction (U = 1722, $p$ = 0.037) were observed, indicating that academics generally value ASNs more highly than SM. The driving forces of homeostasis and selection primarily shape their cognition of ASNs. Homeostasis has a slightly higher value than selection, indicating that ASNs are perceived as more essential for categories such as "related to important cognitive goals", "interest in gaining knowledge about the world", enhancing "analytical cognition", providing "guidance in the search for knowledge", and "advancing one's thought". The selection principle emphasises that academics highly value ASN knowledge content, as it serves for the "development of a higher level of knowledge", "moral development", and "awareness of one's capabilities". Cognition based on the selection principle is a personal, "independent and conscious choice". Both ASNs and SM motivate academics to "revise their ideas" and contribute to "the development of their unique worldview", both of which are essential for "the comprehension of the content". Reduction is less apparent than other principles. However, academics prioritise mental comfort and obtaining knowledge on request, and "efficient, fast and successful cognition" using ASNs rather than SM.

### 5.4. The Epistemological and Sociopsychological Meanings

To address the second sub-question on the epistemological and psychosocial meanings of ASNs and SM and the development of their semantic constructs, the EASQ data were analysed. This analysis involved correlation analysis and exploratory factor analysis, with each source's data cases analysed separately. Initially, the semantic categories of 92 statements were subjected to Pearson correlation to generate a matrix of meanings

(92 × 92). Subsequently, principal component factor analysis was used to process the matrix, reducing meanings into dimensions and creating semantic constructs.

### 5.4.1. Academic Social Networks

The factor analysis of the correlation matrix of ASNs revealed two factors. Among these factors, 46 meanings out of 92 were deemed significant since their factor weights surpassed 0.5. Only the significant meanings were considered to construct the semantic space, and scatter plots were utilised. Figure 5 illustrates the semantic construct of ASNs.

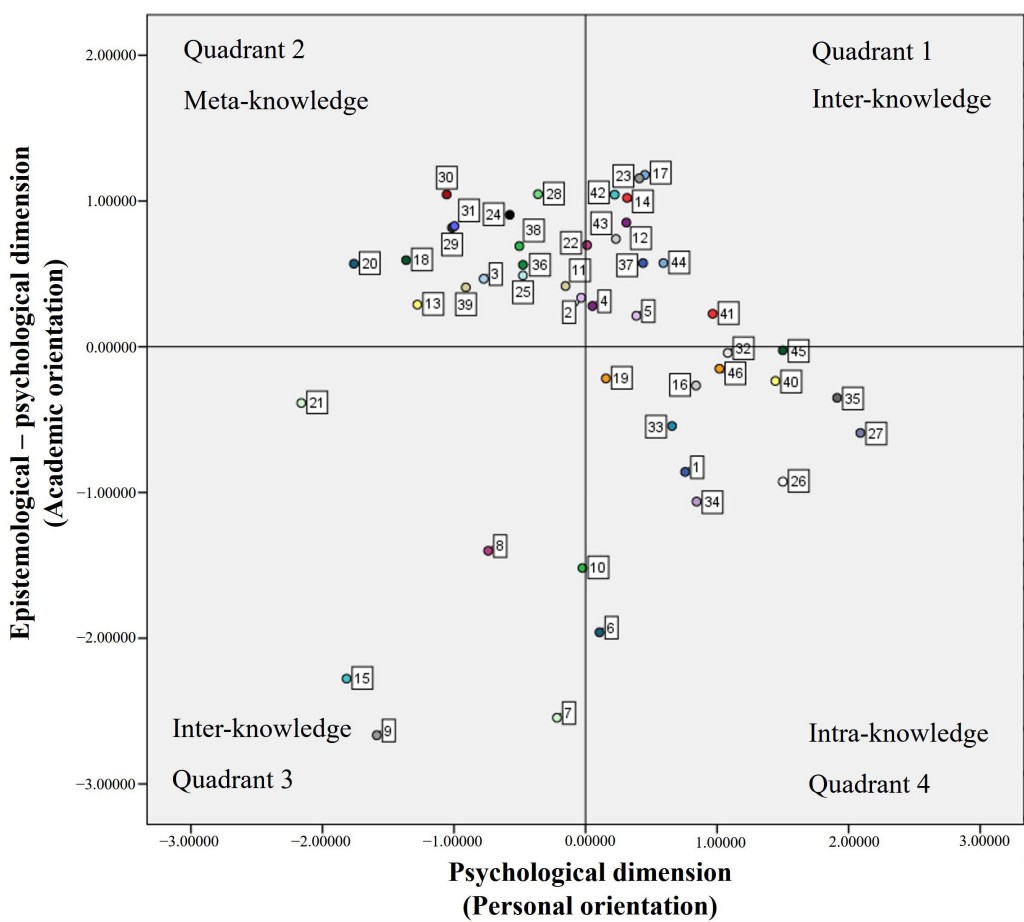

**Figure 5.** The semantic construct of academic social networks.

When analysing the content of these two factors, they were designated names reflecting their semantic categories. The factor represented along the x-axis is known as the psychological dimension, focusing on personal orientation. The y-axis represents the epistemological-psychological dimension, with an emphasis on academic orientation. These two axes create four quadrants where the meanings are located. The interpretation of each quadrant is based on its epistemological and psychosocial meanings, representing the semantic construct of ASNs.

Quadrant 1, named inter-knowledge, encompasses corresponding epistemological and sociopsychological significance within an actual interaction context. These meanings are both essential as ASNs' knowledge and psychosocial value, indicating that the epistemological categories serve as the basis for ASNs usage. Similarly, the psychological meanings further enhance the source's epistemological significance. In essence, academics view these categories as the rationale for valuing and using the source. Academics consider the content of ASNs as the outcome of the interaction of cultural-historical research (17), implying philosophical and theoretical analysis (14) as well as logical and methodological principles (12). This content also incorporates expert knowledge (23) to complete high-level tasks

(42), facilitate intellectual development (43), enhance understanding of our surrounding world (37), and is meaningful and applicable in specific situations (44). Certain categories recognise and find the content relevant, albeit with less emphasis, such as the capacity to address scientific and social problems (5) and the balance between others' and one's own knowledge (41). Other categories hold epistemological value without highlighting personal importance, such as the scientific credibility level (22) and reliable knowledge (4).

Quadrant 2 is known as meta-knowledge. These meanings hold significant epistemological value, but they do not have a direct connection to sociopsychological implications, indicating that the categories do not necessarily correspond in meaning or only slightly do so. They represent an "ideal knowledge" that can prompt development and reflect desired levels of knowledge and cognition. These meanings are crucial in the academic field from epistemological and professional standpoints, though they are less personal. ASNs are used for tasks like organising and managing work within an educational institution (30), conducting research (31), and completing thorough analyses of courses, research, and projects (29). These categories are mostly important with respect to knowledge describing ASNs, such as the result of scientific advancement (24), the means of comprehending knowledge through dialogue (18), and scientific thinking for knowledge creation (20). Nevertheless, some categories are less focused on epistemology and academia, and instead have higher psychological value in terms of personal orientation. This characteristic is evident in categories like cognising the source as valuable in itself (38), stimulating thinking (36), reflecting constructive and creative scientific thinking (25), corresponding to the needs of academic society (3), acknowledging that acquisition requires mental effort (39), and being used for scientific events, conferences, and discussions (28). A category like anthropocentrism (13) is considered personally unnecessary by academics. Additionally, some meanings that hold epistemological value for ASNs but are neutral on a personal level. These meanings refer to fundamental knowledge (11) and the expectation of creating new knowledge (2).

Quadrant 3 is referred to as inter-knowledge and encompasses meanings embodying undesirable epistemological characteristics. These meanings are also personally disinteresting or irritating. It is crucial to recognise the connections between these meanings and establish criteria to avoid or eliminate in social and objective knowledge, to ensure that both the academic and personal cognition of ASNs can benefit from this. It is intriguing to note that within ASNs, the category of new and innovative methodology (15) holds little academic and personal value, akin to non-constructive criticism (9) and the prevalence of beliefs over verified knowledge (8). While the latter two categories seem reasonable, the first category raises some concerns. Yet again, the category that juxtaposes scientific vs. non-scientific knowledge (21) in ASNs is found to be epistemologically more neutral but personally unimportant. Meanwhile, categories such as incompleteness or erroneousness (7), stuck in theoretical positions (6), and the acceptance that truthfulness and falsity of knowledge are equally relative (10) are personally neutral and tolerated. However, from an epistemological standpoint, these meanings carry negative connotations and are better avoided.

Quadrant 4, named intra-knowledge, presents subjective and sociopsychological meanings that may not align with their epistemological and academic value. Despite this, they are applicable in personal cognition with ASNs, even if the epistemological value is low or absent. Personal importance, although lacking in epistemological value, encompasses categories such as elements of religious and metaphysical knowledge (26), sharing knowledge with friends and relatives (34), and the meaning of the studied subject (1). These categories can also be non-academic and non-scientific. Some meanings hold less subjective significance, but their academic and epistemological value is still worth considering. These categories refer to ASNs as applied in new social communication (33), considering the author's evaluation as a reference point (16), and acknowledging that attitudes towards ASNs is the result of socialisation (19). Personal meaningfulness is attributed to ASNs' content that exhibits coherence of scientific and routine rationality (27) and sustained interest (35) per their knowledge value. Some subjectively necessary

categories were identified that vary in their level of concern with academic cognition. These categories indicate that ASNs' content that requires emotional effort (45) is used for supplementing and improving courses/projects (32), fostering new interests (40), and situational, unstructured, or impromptu knowledge acquisition from ASNs (46).

### 5.4.2. Social Media

Through factor analysis, the correlation matrix for the meanings of SM presented two factors. Among the 92 meanings analysed, 39 were found to be significant as their factor weights surpassed 0.5 for the two factors elucidated below. The semantic construct of SM was created by considering only these significant categories, as illustrated in Figure 6.

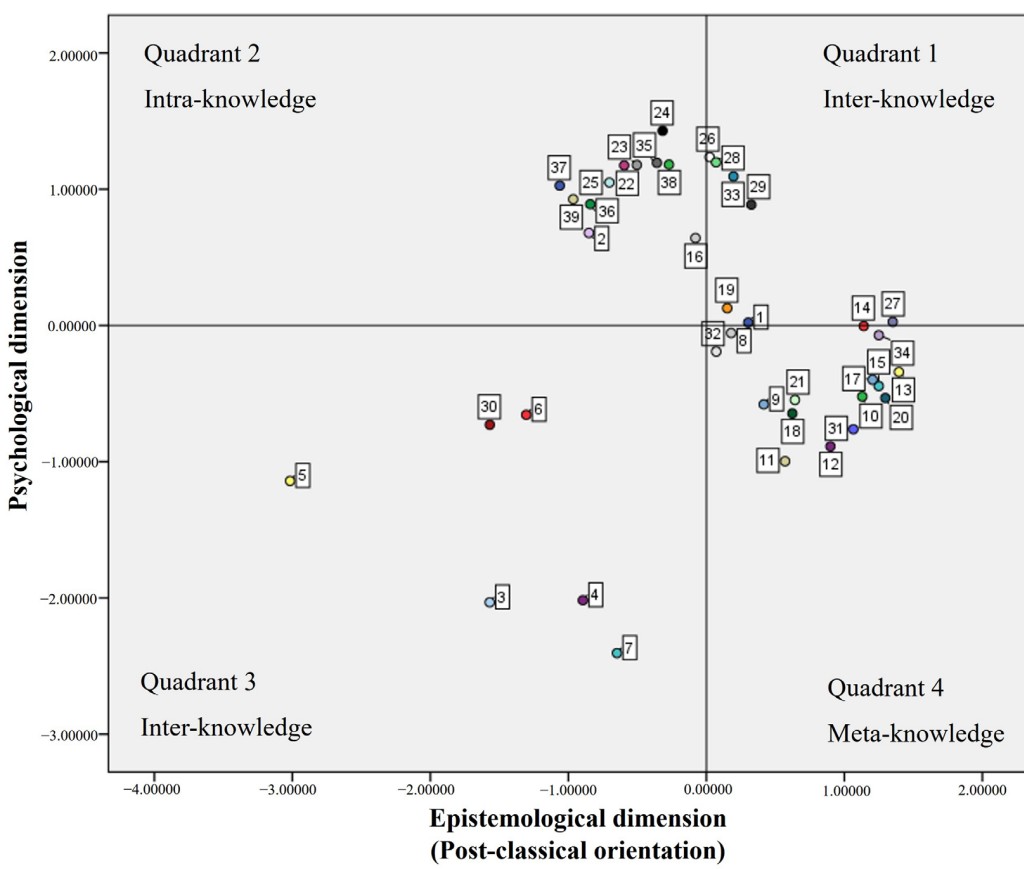

**Figure 6.** The semantic construct of social media.

While analysing the content of the two factors, they were named based on their respective meanings. The x-axis was referred to as the epistemological dimension with post-classical orientation, while the y-axis represented the psychological dimension. The post-classical orientation signifies the re-evaluation of the importance of knowledge in a contemporary context, which we observed as common for both digital platforms while addressing the first sub-question. Four quadrants that correspond to epistemological and psychosocial categories were determined, creating the semantic construct of SM.

Quadrant 1 is referred to as inter-knowledge, characterised by a certain inclination towards epistemological and psychological meanings, though these are not widely expressed, which suggests that not many categories reflect the knowledge aspects of SM with direct psychological implications. However, some categories indicate the subjective importance of SM, and the epistemological value, as not so crucial for solutions for new and uncommon tasks (29) because SM content still fosters new interests (33) and maintains sustained interest (28). It is also used for supplementing and improving study courses/projects (26). Two meanings are implied in both dimensions but not explicitly expressed, including the

expectation of academics for SM content to have a correspondence of the reflected reality to the truth (1) and some scientific credibility level (19).

Quadrant 2, known as intra-knowledge, is characterised by primarily subjective and sociopsychological meanings. These meanings are valued for their pragmatic use and are not concerned with the objective value of knowledge as supported by the categories that indicate that SM are used for sharing knowledge with students and colleagues (24). The importance of SM in this context is primarily behavioural as it requires emotional effort (38) and is applicable to organising and managing work within an educational institution (23) and completing tasks at a high level (35). Interestingly, some subjective meanings still appraise the practical value of knowledge in an epistemological sense. These categories define SM as meaningful with applicability in specific situations (37), for doing research (25), developing one's scientific thinking (22), having personal meaning (39), serving for intellectual development (36), and expecting to gain describable and explicable aspects of reality (2). We found it intriguing that academics value subjective and epistemological competencies possessed by SM authors, such as admitting mistakes and shortcomings (16) in their content.

Quadrant 3 is defined as inter-knowledge, and encompasses meanings that reflect undesirable epistemological features and personal disinterest. These meanings highlight the need to avoid certain aspects of SM for both reasons mentioned above. They emphasise the prevalence of beliefs over verified knowledge (6), the omission of controversial aspects (5), and biased interpretations (3). Interestingly, even content that promotes avoidance is considered thought-provoking (30). While some categories may be personally unacceptable and subjected to criticism, they are somewhat understandable from a post-classical perspective. They suggest that SM content may be incomplete or erroneous (4) and may contain replicated knowledge (7).

Quadrant 4 is referred to as meta-knowledge, containing meanings that hold high epistemological value when viewed through a post-classical lens. They represent the "ultimate wisdom" that SM should strive to provide, leveraging digital media capabilities to offer qualitative knowledge. These criteria do not directly relate to the importance of personal interaction with SM as is evident in categories that are crucial for SM to possess from an epistemological standpoint. SM need to obtain such attributes as scientific tolerance and pluralism (20) and various cognitive approaches (10). These meanings also imply that the author's evaluation serves as a reference point (13) and highlight the importance of comprehension through dialogue (17) and collaboration (15). The epistemological categories are also present in the actual context of SM, as subjective meaning is less negative and objective value is recognised. SM provide comprehension as the process of constructing meanings (12), helping in understanding the world around us (31), where the author integrates their own thinking and experience (11). The mindset towards SM is recognised as the influence of social and meta-knowledge-categorised attitudes resulting from socialisation (18). SM have the potential to translate more objective and transcendental understanding like contemporary ideals, values, and worldviews (9), and incorporate elements of religious and metaphysical knowledge in scientific knowledge (21). Some socially epistemological applications are also expressed, such as applying in new social communication (27) and acknowledging the interaction of the authors' cultural-historical research (14) in creating knowledge. SM offer a balance between other people's and one's own knowledge (34). Eventually, neutral but present meanings are observed that reflect that SM also contain philosophical and theoretical analysis (8) and encourage self-reflection (38).

## 6. Discussion

The emergence of an informational and interactive paradigm in today's academic landscape has resulted in digital media playing a vital role in cognition, communication, and information dissemination. As a result, academics are now proficient users and audience members, listeners, readers, and communicators, as well as creators of digital content on academic social networks (ASNs) and social media (SM). Considering this, we sought

to delve into the epistemological and psychosocial aspects of these knowledge sources from the perspective of academics through this study. Drawing upon the epistemological attitude (EA) model framework, we strived to address the need for research on knowledge encompassing ASNs and SM.

In this essay, we examined two research questions that shed light on knowledge quality, prognosis, use, and worth. To address our first question, we examined how academics understand, approach, use, and value ASNs and SM as reliable sources of knowledge. Our findings generally confirm the perceived importance of ASNs and SM as valuable sources for knowledge acquisition, creation, sharing, and communication for two primary purposes: academic and personal. The significance of these platforms has been scientifically confirmed and ecologically valid [1].

The results divulged the distinct attributes of academics' EA towards ASNs. Overall, the content of ASNs is regarded as acceptable and up-to-date, but its limitations in terms of knowledge potential and required verification are also recognised, highlighting both optimism and scepticism. Academics approach and address knowledge issues through individual and collaborative methods, using a particular lens to understand the content. The rejection of subject-centrism and the inclination towards subject-centrism are emphasised. Additionally, academics adhering to fundamentalism and normativism acknowledge that ASN content aligns with high quality, theory, and methodology. Interaction with ASNs is preferred within the academic context, where research and education are primary functions. The motivation to engage with ASNs and the value placed on the knowledge is manifested through the homeostasis principle, which highlights academics' functional adaptation in obtaining and using practical knowledge. It is also shown that ASNs contribute to the social capital of academics by enabling self-promotion, providing professional knowledge, and facilitating interaction within a peer community [4,13].

Academics' EA towards SM also revealed the particularities of their approach and the significance they attributed to knowledge. Their epistemological strategy is prominently expressed through scepticism, which acknowledges doubts about the potential and quality of knowledge. Agnosticism is also a defining feature, as academics recognise the impossibility of SM attaining a complete comprehension of reality and acquiring trustworthy knowledge. The epistemological approach of academics is characterised by criticism, emphasising the difficulty of distinguishing between knowledge and subjective beliefs. The role of individuality in understanding and providing knowledge is highlighted by subject-centrism and the rejection of subject-centrism, as observed in the use of profiles and posting options on SM, as well as the collaborative and innovative nature of knowledge co-creation in the current milieu. This aligns with the idea that scholars express their individuality within SM [11], appreciate posting performance [15], and collaboratively create and manage content [18]. The rejection of fundamentalism and normativism, as well as the rejection of science-centrism, reflects that a lack of solid theoretical and scientific grounds is attributed to SM. Academics' interaction with SM is primarily influenced by personal context rather than academic context. In line with this interaction, the motivation that determines the cognition of SM is reduction. Academics' motives for employing SM are primarily for leisure and without mental effort. This tendency to use SM is linked to sources of reward, such as passing the time and showing affection [40].

A comparative analysis of academics' EAs towards ASNs and SM revealed academics' distinct orientations towards these sources. They display a prominent sense of optimism towards ASNs, while scepticism and agnosticism are more characteristic towards SM. Academics approach knowledge based on its objective value and definitions, employing different methods to identify and solve knowledge problems. Scholars underscore the importance of using a critical approach to explore knowledge in SM. They exhibit a higher degree of subject-centrism in SM platforms than ASNs. Contrarily, academics perceive knowledge in ASNs as relying on fundamentalism and normativism, with support from a science-centric perspective. Overall, academics share a post-classical epistemological approach to digital content, characterised by post criticism and the rejection of fundamen-

talism and normativism, subject-centrism, and science-centrism in both knowledge sources. ASNs are used more frequently and extensively in both professional and personal contexts, and scholars use them more often in academic settings. In personal, interpersonal, and social situations, both sources are equally applicable and less intensive. These findings confirm that ASNs are preferred by academics for both scholarly and private communication [3,4] and in general, academics assign a higher value to ASNs compared to SM. The main driving forces directed to ASN application are balanced and selective cognition principles.

This comparison between ASNs and SM highlights the differences in their knowledge issues. ASNs are generally considered a more trusted, credible, and functional source within academia. Despite being a media source, ASN content primarily consists of scientific writing and other scientific and pedagogical material. However, academics still exercise caution and scrutiny when perceiving and analysing their content. Again, SM are perceived more as a source of information rather than a reliable source of knowledge. This information is subject to criticism, and academics are generally mindful of the potential pitfalls of excessive SM usage.

In the second question, we explored and addressed the significance of both sources of knowledge by creating semantic constructs. Our findings revealed that each knowledge source presents distinct epistemological and psychological meanings. These results emphasise the importance of ASNs in present-day academic interaction and research [1]. Furthermore, we gained insights into the general knowledge contained within ASNs as a source type. The semantic construct of ASNs revealed two dimensions, and it is crucial to analyse their interaction. The dimensions were classified as the epistemological-psychological dimension with an academic orientation and the psychological dimension with a personal orientation.

Our findings are consistent with real-life situations and, most importantly, they support the EA concept, which posits cognition occurring at subjective, contextual, and epistemological levels, encompassing intra-, inter-, and meta-knowledge. We identified these levels within the semantic constructs' dimensions and synthesised the semantic categories' organisation, which led to identifying four spaces or quadrants: two spaces representing inter-knowledge and two spaces representing intra- and meta-knowledge.

Inter-knowledge encompasses the epistemological and sociopsychological meanings practised by scholars in both academic and personal contexts. The meanings attributed to ASNs highlight their objective and scientific knowledge that are highly valued and utilised for valid reasons. Studies have also shown that ASNs are highly regarded for their ability to provide credible knowledge [13]. Interestingly, there are situations where opposing the inter-knowledge dimension also arises, reflecting categories that serve as objective criteria to determine whether to use certain content. These undesired epistemological features are personally disinteresting. It is intriguing to note that qualities such as non-constructive criticism, the prevalence of beliefs over verified knowledge, and the use of new and innovative methods are perceived as undesirable from both perspectives. This observation may be related to ASNs allowing authors to share their work without going through the traditional editorial board and review process. As a result, academics have specific requirements for ASN content, which scientists should acknowledge. Innovative knowledge is expressed to be proven in this context.

The intra-knowledge space is characterised by subjective and sociopsychological meanings that do not consider their correspondence with epistemological and academic values. This indicates that ASNs also apply to personal cognition, even if the epistemological value is absent or low. This refers to sharing content, establishing communication bases, and sustaining interest by obtaining information spontaneously.

The space of meta-knowledge reflects the idea of "ideal knowledge" that ASNs should contain. Also, it encompasses the motivations for desired knowledge, levels of understanding, and epistemological criteria only applicable within the academic context, scientific cognition, and research. This dimension demonstrates that ASNs are recognised as containing

objective knowledge and a meta-sphere where scientific, philosophical, and metaphysical understanding can occur.

The semantic framework of SM also presents two dimensions, but with different orientations in knowledge, as reflected in their respective value. We have identified two dimensions: the epistemological dimension with a post-classical orientation and the psychological dimension. The psychological importance of SM supports prior research that underscores personal and social motivations for their utilisation [38,39]. However, the knowledge issue of SM is a cause for concern and is a subject that demands greater awareness in academia and society. The post-classical orientation refers to the revaluation of the importance of knowledge within a contemporary context. This orientation is commonly observed in comprehending digital content, as observed in our response to the first question. Interpreting knowledge qualities sheds light on certain trends within SM content.

Our findings have revealed two interconnected semantic spaces of knowledge. The first space of inter-knowledge indicates the direct and coherent psychological implications of only a few knowledge qualities in SM. This space acknowledges the subjective importance of SM and the objective value of knowledge, while the latter is deemed less relevant as is consistent with previous findings that emphasise the interplay between the actuality of cognition, communication, and real-world events [2], as well as the sharing of experiences, the expression of knowledge, and the generation of ideas [36].

The second space of inter-knowledge, similar to ASNs, does not possess desired epistemological features and lacks elements for inducing personal interest. However, this dimension surprisingly exhibits consistent regularities, suggesting that even SM content encompasses knowledge based on beliefs, controversies, and biased interpretations, although academics avoid an excessive contemplation of these aspects. When SM are considered untrustworthy and suspicious, there is no reason to consider them, as it confirms that society tends to be polarised towards SM content [28].

The intra-knowledge dimension indicates that SM primarily hold subjective and sociopsychological meanings due to the pragmatism and ignorance of their objective value. Other studies have also revealed that personal and interpersonal motives take priority in using SM [44]. Lastly, the meta-knowledge space, as categorised by academics, has shown that SM are perceived as a potential platform for developing knowledge. Academics expect that SM may have epistemological importance and capture the post-classical value of knowledge, indicating that SM the potential to serve as a means of objectifying and verifying knowledge. SM could foster scientific plurality, acceptance, and the co-existence of different paradigms, potentially transcending social expertise. Although academics have yet to witness the widespread application and realisation of desired meta-knowledge through SM, there is a belief that SM content may become more sophisticated in the future. Researchers also acknowledge the benefits of SM in knowledge transfer [23], as their quality is constructed in real-time. We agree that SM hold potential, given their significant entrepreneurial and functional role in other professional spheres [21].

The limitations of this study lie in its theoretical and methodological novelty. Further research is needed to expand the EA concept and validate its model using specific sources. Additionally, the EA method requires further validation, and the scales must be refined. Conducting studies with academicians from different countries and larger sample sizes is also important. These studies should use improved versions of both the EAQ and EASQ questionaries.

The study's practical implications and applications can be seen from both a theoretical-methodological and practical perspective. From a theoretical-methodological standpoint, the development of the cognition system theory has the potential to explore the subject and object relations and subject–knowledge relations. The latter incorporates multi-level reflection. Furthermore, it captures the relations between objects and knowledge, encompassing cognition as a synergistic amalgamation of multiple systems and their interconnections.

The methodological aspect involves developing the EAQ and EASQ methods to investigate knowledge regarding knowledge and value. This has direct relevance to

practical applications, particularly in the fields of education and social cognition. The EA methodology provides a comprehensive perspective for studying knowledge sources, which is crucial in humanities and social sciences research. By creating constructs of meanings for specific sources and differentiating between types of sources, we can gain knowledge and meta-knowledge about these sources.

**Author Contributions:** Conceptualization, J.S.; methodology, J.S. and A.V.; software, J.S. and V.R.; validation, J.S. and V.R.; formal analysis, J.S. and V.R.; investigation, J.S.; writing—original draft preparation, J.S.; writing—review and editing, J.S. and A.V.; visualization, J.S.; supervision, A.V.; project administration, J.S.; funding acquisition, J.S. and A.V. All authors have read and agreed to the published version of the manuscript.

**Funding:** This research was funded by the ESF Project No. 8.2.2.0/20/I/003 "Strengthening of Professional Competence of Daugavpils University Academic Personnel of Strategic Specialisation Branches 3rd Call". Nr. 14-85/14-2022/10.

**Informed Consent Statement:** Informed consent was obtained from all subjects involved in the study.

**Data Availability Statement:** The data presented in this study are available upon reasonable request from the corresponding author owing to the continuing development of the methodology.

**Conflicts of Interest:** The authors declare no conflict of interest.

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
