# Peer review of "Academics’ Epistemological Attitudes towards Academic Social Networks and Social Media"

_philosophies, doi:10.3390/philosophies9010018_

Round 1

Reviewer 1 Report

Comments and Suggestions for Authors

The article “Academics’ Epistemological Attitudes Towards Academic Social Networks and Social Media” addresses a current topic, which is relevant and has practical repercussions.

For its publication, it would be advisable to make a series of changes, which affect the structure and content. In this regard, the comments here follow the order of the text received.

(i) The first pages are dedicated to the common and differentiating features of Social Networks and Academic Social Networks. Thus, instead of “Introduction”, which is what is usually put when someone doesn't know what to put, a title with content would be more appropriate, such as the following: Academic Social Networks and Social Media: Common Features and Differences

By changing the title, a more systematic and less long text in section 1 would be easier. It would make reading more fluid and with fewer words. Putting two subheadings, one for common features and another for differences, would make the reading clearer and more systematic.

(ii) On page 4 the central question of the article is posed:
“we formulated the main research question: What are the epistemological and psychological aspects of ANSs and SM from the perspective of academics?”

Obviously, the question to be investigated should appear before in the text of the article, since all research is related to solving a problem, which gives rise to a question or several that are relevant to the topic at stake.

(iii) The text has a strange structure, since the methodology appears at the end of everything (section 5), while the results are offered in section 3. This does not seem appropriate, since the results depend on the methodology used.

Therefore, a more reasonable structure of the article would be the following:

1. Academic Social Networks and Social Media: Common Features and Differences (or an equivalent title, to reflect the content of this section).

2. Epistemological Attitude Theory.
3. Methods and Materials (instead of “Materials and Methods” and, in any case, before the results, that is, as section 3 and not as section 5).
4. Results (now it is number 3, but the logical thing is that it comes after the methodology).
5. Discussion (now it is number 4).

(iv) A terminological review would be advisable in a series of cases, such as the following:

Page 5. Transcendental knowledge means “all knowledge from all possible perspectives”, when instead of global or all-encompassing knowledge, transcendental knowledge is something that transcends the given, in that it is beyond a given reality. It is not something cumulative but of a different
order.

Page 6. The epistemological dimension includes the subjective dimension and the contextual (or intersubjective) dimension. Then the objective dimension can be added.

Page 8. “Homeostasis” is related to self-regulation. In principle, you do not “define the knowledge value”, which is the axiology of research.

Page 9. “Agnosticism” has more to do with the “unattainable” than with “deny” a source to understand reality, which would be a version of skepticism.

Page 11. “Semantic constructs” is about language. But, in the text, these are rather intelligible constructs and, certainly, these constructs are not reducible to language. In fact, language, knowledge, and thought are constantly interrelated in the text.

Pages 16-17. There is a 42-line paragraph. This does not favor the reading of the text in any way.

(vi) In general, the text needs more systematicity. Sections 1 and 4 do not have any subheadings, while section 5 has a radically different presentation, full of sections.

A publishable version requires major changes, first of a structural nature and, second, of conceptual clarity in the analysis. The validity of the conclusions depends on knowing how to value and evaluate the different responses received. This requires rethinking the content and then presenting it in
a more systematic and coherent way.

Comments on the Quality of English Language

Moderate editing of language is required.

Author Response

Dear Reviewer,

We deeply appreciate your profound interest and the value of our work. Thank you for your constructive comments. We thoroughly articulate our point-by-point enhancements, coherence, and feedback.

The changes suggested in your comments are highlighted in cyan in the manuscript.

Best regards, authors.

Reviewer 2 Report

Comments and Suggestions for Authors
  1. While the authors' optimistic analysis of social networks in academia is commendable, it overlooks critical shortcomings that warrant careful consideration. If you could add some of them in this paragraph on p. 3 "The widespread use of social media has become common, often leading to positive 107 outcomes but sometimes resulting in less desirable effects [22]. The studies show that SM 108 can have adverse effects, including increased distractions and disturbances [23] and de-109 pressive symptoms [24]. Overusing social media has been linked to various negative con-110 sequences, such as decreased productivity [25]. Because of that, we also can observe the 111 polarisation of attitudes towards SM in society [26]. " it would be great.

  2. There is a genuine concern about inadvertently legitimizing social networks in academia through articles that overly highlight their positive aspects, potentially overshadowing the significant challenges and ethical considerations associated with their widespread use.

Author Response

Dear Reviewer,

Thank you wholeheartedly for your constructive and positive review. We admit that we have not adequately demonstrated the adverse effects of social media. We offer additional evidence and arguments to present a comprehensive view of social media.

Best wishes, authors

Round 2

Reviewer 1 Report

Comments and Suggestions for Authors

The text has undergone a thorough revision, so that it has improved appreciably. This makes it now publishable. But it is worth paying attention to some details, such as that the discussion cannot be number 4. With the new structure it has to be number 6.

Author Response

Dear Reviewer,

Thank you very much for your feedback. We express our gratitude for your guidance in directing us towards the relevant details in both reviews.

The changes made during this revision phase are marked in magenta colour for easy identification. We carefully checked the numbering of the sections. References have been reordered, unnecessary ones excluded, grammar and language checked, and repeated phrases and sentences deleted. These changes can be visible via the track changes option.

We have entered the year 2024, and we send our best wishes for success in all your undertakings.

Best wishes, Authors